# Psychotropic deprescribing across different prescribing professions in New Mexico and Louisiana

Phillip M. Hughes[1,2]*, Joshua D. Niznik[1,3,4], Robert E. McGrath[5], Casey R. Tak[6], Robert B. Christian[7,8], Betsy L. Sleath[1,2], Kathleen C. Thomas[1,2]

**1** Division of Pharmaceutical Outcomes and Policy, Eshelman School of Pharmacy, University of North Carolina, Chapel Hill, North Carolina, United States of America, **2** Cecil G. Sheps Center for Health Services Research, University of North Carolina at Chapel Hill, Chapel Hill, North Carolina, United States of America, **3** Division of Geriatric Medicine and Center for Aging and Health, School of Medicine, University of North Carolina at Chapel Hill, North Carolina, United States of America, **4** Center for Health Equity Research and Promotion, Veterans Affairs (VA) Pittsburgh Healthcare System, Pittsburgh, Pennsylvania, United States of America, **5** School of Psychology and Counseling, Fairleigh Dickinson University, Teaneck, New Jersey, United States of America, **6** Department of Pharmacotherapy, University of Utah College of Pharmacy, Salt Lake City, Utah, United States of America, **7** Carolina Institute for Developmental Disabilities, University of North Carolina at Chapel Hill, North Carolina, United States of America, **8** Department of Psychiatry, School of Medicine, University of North Carolina at Chapel Hill, North Carolina, United States of America

* phughes1@email.unc.edu

## Abstract

This study aimed to assess differences in the rates of deprescribing between prescribing psychologists, psychiatrists, and primary care physicians. MarketScan private insurance claims were used to develop a longitudinal active-comparator, prevalent-user cohort of patients who were treated with a psychotropic medication from psychologists, psychiatrists, or primary care physicians for at least 90 days in New Mexico or Louisiana (states where psychologists can prescribe) between 2005–2021. The type of provider (psychologist, psychiatrist, or primary care physicians) who prescribed the psychotropic medication was the exposure of interest. Three measures of deprescribing were used as outcomes: deprescribing without replacement, complete discontinuation of prescribing, or a sustained reduction in the prescribed days' supply. Patient demographic and clinical characteristics during the six months prior to their initial prescription were measured as covariates. Inverse propensity of treatment weighting was used to adjust for baseline differences between provider groups, creating two weighted analytic cohorts with covariate balance: psychologists versus psychiatrists and psychologists versus primary care physicians. We estimated doubly-robust Cox Proportional Hazards models for each deprescribing measure in both cohorts. Prescribing psychologists deprescribed without replacement more than psychiatrists (Hazard ratio [95% CI] = 1.13 [1.06, 1.20]) and less than primary care physicians (0.73 [0.69, 0.78]). Conversely, they reduced the days'

**Data availability statement:** The data used in this study (MarketScan Commercial Claims and Encounters Database) is proprietary data owned by a third-party vendor and may be accessed by purchasing the data from the vendor (Merative). Information about how to purchase access to the database can be found here: https://www.merative.com/documents/merative-marketscan-research-databases. The authors has no special access privileges that others who purchase the data would not have.

**Funding:** This research was partially supported by a National Research Service Award Pre-Doctoral/Post-Doctoral Traineeship from the Agency for Healthcare Research and Quality sponsored by The Cecil G. Sheps Center for Health Services Research, The University of North Carolina at Chapel Hill (T32-HS000032 to PH). The database infrastructure used for this project was funded by the Department of Epidemiology, UNC Gillings School of Global Public Health; the Cecil G. Sheps Center for Health Services Research, UNC; the CER Strategic Initiative of UNC's Clinical Translational Science Award (UL1TR001111); and the UNC School of Medicine. The funder played no role in the planning, conducting, writing, or decision to publish the study.

**Competing interests:** Phillip Hughes was awarded the 2023 Patrick H. DeLeon Prize for Outstanding Student Contribution to the Advancement of Pharmacotherapy from APA Division 55 (Society for Prescribing Psychology). Both Phillip Hughes (2024) and Robert McGrath (2021) have received the John D. Preston Award for Outstanding Contributions to Clinical Psychopharmacology from APA Division 55 (Society for Prescribing Psychology). The remaining authors have declared that no competing interests exist.

supply less often than psychiatrists (0.79 [0.69, 0.91]) and more than primary care physicians (1.64 [1.42, 1.90]). There were no differences in complete discontinuation between provider types. Prescribing psychologists deprescribe at a rate between psychiatrists and primary care physicians. Findings varied depending on the deprescribing measure used, suggesting psychotropic-specific deprescribing measures are needed.

## Introduction

Approximately 1 in 7 (13.9%) people received a psychotropic medication for their mental health in 2021, including 36.1% of those with a mental illness [1]. There is limited research to date on the long-term utility and safety of psychotropic medications, and the evidence available provides mixed results. Long-term treatment with antidepressants, for example, does not appear to provide protection against future depressive episodes and may increase risk in some instances [2]. Similarly, a 5-year study of psychotropic medication use in patients with schizophrenia found that while antidepressant and antipsychotic use lower mortality rates, long-term use of mood stabilizers was actually associated with higher mortality rates [3]. In the face of limited, often conflicting evidence regarding the long-term safety of psychotropic medications, there has been a growing emphasis on "deprescribing," or intentionally stopping/tapering patients off of, psychotropic medications [4–6].

Psychotropic deprescribing is an emerging area of research, and the literature to date has primarily focused on efforts to reduce the burden of complex medication regimens and adverse events (e.g., delirium, falls) among the geriatric population [7–10]. However, it is well established that many psychotropic medications can have burdensome side-effects for populations of all ages (e.g., weight gain, lowered sex drive, sleep disturbance), suggesting that research on the clinical practice of deprescribing psychotropics should be extended to include other age groups [11]. A small number of studies has focused on barriers and facilitators to psychotropic deprescribing in the mental health context [5,12]. These studies describe patient-provider partnerships as the key facilitator to successful psychotropic deprescribing, while myriad barriers exist, including a lack of formalized guidelines, challenging withdrawal symptoms, and concerns about causing harm or undoing progress [5,12]. Given that the barriers in particular focus on the technical challenges of deprescribing, different types of prescribers (e.g., primary care vs mental health specialist) could have differing levels of comfort with deprescribing, which may translate into different rates of deprescribing across different prescribing professions.

The potential for different deprescribing rates by prescriber type is of particular interest in the United States, as there is a long-standing shortage of mental health specialists [13,14]. This has resulted in the majority (approximately 60%) of psychotropic medications being prescribed in primary care, predominantly by general practitioners (43.5%) [15]. Conversely, mental health specialists, including psychiatrists (33.5%) and prescribing psychologists (2.2%), account for the minority of prescribing.

Taken in this context, if the barriers to deprescribing described in the literature primarily impact primary care physicians (PCPs) rather than psychiatrists or prescribing psychologists, then this would impact the majority of individuals receiving psychotropic medications and therefore warrant further examination. However, the rates of deprescribing by these different prescriber groups have not yet been described.

In contrast to PCPs, who may encounter disproportionate barriers to deprescribing, psychologists are well-positioned to deprescribe due to their training and emphasis on the therapeutic relationship. In response to mental health workforce shortages, seven states have created a pathway for doctorate-level psychologists to seek prescriptive licensure upon completion of additional education in psychopharmacology, with New Mexico and Louisiana being the first to do so [16,17]. Advocates of prescriptive authority for psychologists (RxP) argue that prescribing psychologists have a role to play in reducing unnecessary psychotropic prescribing, often summarized as "the power to prescribe is the power to unprescribe [deprescribe]" [16,18]. As a result, deprescribing is considered a core component of prescribing psychologists' approach to patient care, and the discomfort with deprescribing reported by other prescribers is therefore less likely to manifest in prescribing psychologists. A national survey of prescribing psychologists supports this, finding that prescribing psychologists decreased the dosage or removed a medication entirely for approximately 30% of their patients [19]. Furthermore, given that the therapeutic alliance between patient and provider is a key component of psychotherapy, it is possible that prescribing psychologists fully benefit from the role of this relationship in facilitating successful deprescribing. It has not yet been demonstrated how this rate of deprescribing compares to the rates for PCPs and psychiatrists, however, obfuscating whether prescribing psychologists are characterized by a tendency to deprescribe psychotropic medications more often than their peers.

Given the increasing focus on deprescribing, the unexplored possibility for known deprescribing barriers to impact prescribers differently, and the pro-RxP argument that prescribing psychologists will deprescribe more, the present study examined differences in the rates of deprescribing between prescribing psychologists, psychiatrists, and PCPs. Given the focus prescribing psychologists place on deprescribing, we hypothesized that they will deprescribe medications more frequently than psychiatrists or PCPs. Finally, while deprescribing is considered a positive outcome in the context of this study, we also acknowledge that long-term medication usage may be the correct treatment decision for a given patient (e.g., antipsychotics for youth with schizophrenia) [3].

## Methods

### Study design & sample

The present study expands on a previously described active comparator, new user longitudinal cohort study designed to estimate the effect of prescriber type on patient outcomes in New Mexico and Louisiana [20]. Using Merative MarketScan data from 2005-2021, the previous study identified a cohort of new users of psychotropic medications prescribed by psychologists, psychiatrists, and PCPs. Those in the original cohort were required to be continuously enrolled in their private employer-sponsored insurance plan (including mental health and prescription coverage) for at least 6 months prior to the initial psychotropic prescription. The date of their first psychotropic medication prescription claim was used as the date of treatment initiation.

To assess deprescribing in the present study, establishing a cohort of patients with consistent psychotropic use was necessary in order to identify changes in prescribing that could be indicative of deprescribing. To accomplish this, individuals from the initial cohort were retained if they had psychotropic medication on-hand 90 days after treatment initiation or did not have their final psychotropic prescription during the 90-day initiation window. Additionally, we excluded individuals who disenrolled from their insurance plan during the initiation period. The result is an active comparator, prevalent user design nested within a new user design. In using this approach, the benefits of the active comparator component (i.e., protection against selection bias due to frailty or contraindications) and new user component (i.e., a uniform treatment starting point and temporal ordering of covariates, treatments, and outcomes) are retained while offering the ability to

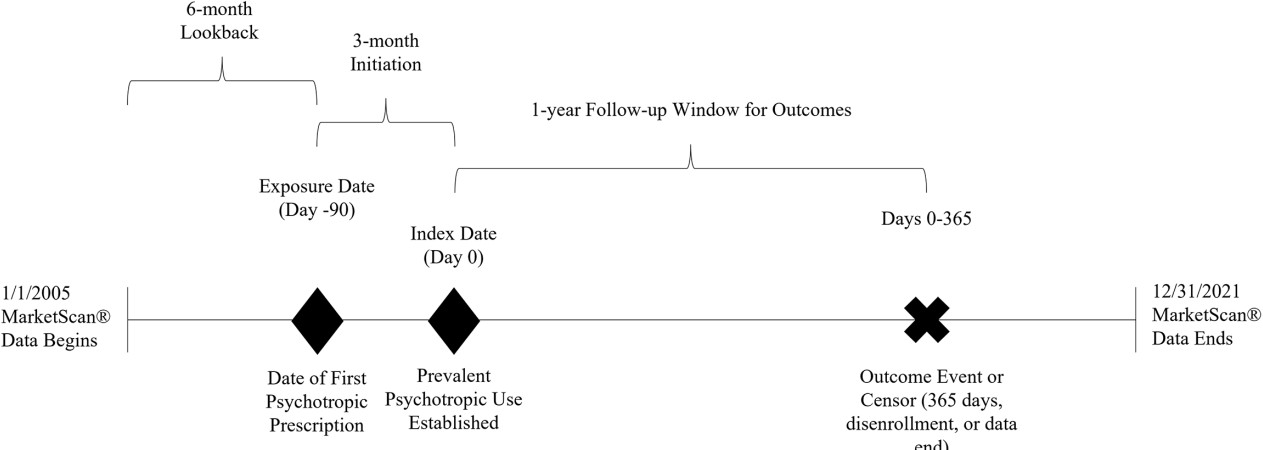

**Fig 1. Active comparator, prevalent user longitudinal cohort nested within a new user cohort.** Legend: "Index Date" refers to the date of the patient's first prescription of a psychotropic medication. Patients must have been continuously enrolled (including mental health and prescription coverage) during the 6-month lookback period to be as certain as possible that the index medication was the first. Patients must also have been continuously enrolled during the 90-day initiation period. Individuals who had no medication on hand at day 90 or had their final prescription prior to day 90 were excluded. The active comparator component of the design is achieved through the different prescriber types (prescribing psychologists, psychiatrists, and primary care physicians) at index.

study outcomes specific to prevalent users (Fig 1) [21]. This approach yielded a cohort of 139,119 individuals in who were continuously adherent for at least 90 days on a psychotropic medication prescribed by a prescribing psychologist, psychiatrist, or a PCP.

Patients who were retained in the cohort were assigned a new index date exactly 90 days after their treatment assignment (initial prescription claim) and deprescribing outcomes were assessed for up to a year (365 days) after the index date. The use of a 90-day initiation window allows us to focus on deprescribing that is not due to turbulence during the initial medication initiation period (e.g., stopping a medication during the initial prescription due to side-effects). A window of 90 days was selected through consultation with the psychologist and psychiatrist on the study team, as well as being a common initiation window used in establishing polypharmacy (concurrent use of multiple medications) to accommodate tapering schedules for certain psychotropic medications [22]. Patients who did not experience deprescribing during the follow-up were considered censored at the end of the year or their last day of enrollment. This study was reviewed and deemed exempt from oversight by the University of North Carolina at Chapel Hill Institutional Review Board. As an analysis of secondary data, informed consent was neither required nor obtained. Study data were accessed on 10/21/22, and no member of the study team had access to identifiable information during or after data collection.

## Measures

*Prescriber Type.* The type of prescriber who initiated the patient's psychotropic medication is not identified in the MarketScan database. We instead relied on an algorithmic approach described in prior studies using this cohort [20,23]. This approach examined the various providers with whom the patient had a billed visit in an outpatient or inpatient settings in the 30 days prior to the initial psychotropic prescription. If they ever had a billed visit for a psychiatrist, it was assumed that the prescription came from the psychiatrist. If not, but they did see a PCP (family medicine, internal medicine, or pediatrics), the prescription was assigned to that provider. Finally, if they saw neither a psychiatrist nor PCP but saw a psychologist, the prescription was assigned to the psychologist. Using this algorithm, visits with advanced practice providers (such as nurse practitioners and physician associates) were not examined in this study such that patients who did not see

a psychiatrist, PCP, or psychologist were excluded. For patients who initially received a prescription from a PCP but were subsequently referred to either a psychiatrist or prescribing psychologist, their prescription would only be attributed to the specialist provider if the specialist appointment was billed prior to the initial prescription fill. If the specialist appointment occurred after the initial prescription, then the PCP would be identified as the prescriber. Although this scenario is not unlikely, prior research with this cohort demonstrates that the PCP cohort is markedly different from the specialist cohorts, suggesting this may not be a significant problem [23].

Two alternative definitions were considered in sensitivity analyses. First, we examined a conservative approach in which a psychologist was assigned as the prescriber only if the patient also saw no other prescribers (e.g., no nurse practitioners) during the 30 days (the "strict hierarchy" definition). Second, we used the less stringent but most commonly used approach in which the prescriber was identified as the provider for the most recent claim prior to the prescription claim (the "recent provider" definition) [24,25].

*Deprescribing.* Defining deprescribing in claims data is a challenging and relatively unexplored area [26]. For this reason, we used three definitions of psychotropic deprescribing: 1) deprescribing without replacement, 2) complete discontinuation of prescribing, and 3) a sustained reduction in the prescribed days' supply for a medication. These approaches are commonly used in the literature on deprescribing, with the first two being different measurements of the construct "discontinuation" and the third measuring the construct "deintensification" [26]. The first approach, deprescribing without replacement, was defined as a 30-day lapse in medication supply for a given medication (e.g., a 14-day prescription would be considered deprescribed 44 days after the prescribing date). Additional medication supply lapse lengths of 90 and 180 days were examined in sensitivity analyses. To assess replacement, any new class of psychotropic medication (e.g., an antidepressant switching to an anxiolytic) prescribed during the lapse period defining deprescribing was considered a replacement. For the second approach, complete discontinuation of prescribing was identified as the date of the last psychotropic prescription plus the days' supply for the prescription (e.g., a final prescription on day 60 for a 30-day supply would be defined as deprescribed on day 90). Finally, we measured deprescribing in the form of deintensification via any reduction in days' supply that was sustained for two consecutive prescription claims. We were unable to examine changes in daily dose directly given that daily dosage is drug specific and could vary substantially within drug class. To examine the impact of minor fluctuations in days' supply, we conducted a sensitivity analysis in which we required a sustained reduction of at least 7 days' supply (e.g., a 90-day prescription followed by two 85-day prescriptions would not be considered a reduction under this definition). For all three approaches, patients with a days' supply with illogical values (e.g., 999 days' supply) were censored at the date of the illogical value. Final prescriptions occurring outside the 365-day follow-up window were censored at day 365.

This approach to assessing deprescribing is intentionally broad given that our focus is on general deprescribing patterns as a clinical practice behavior of different providers types. While using this approach allows us to most succinctly compare these broad practice behaviors, we acknowledge that the clinical decision to deprescribe is highly dependent upon the individual patient and medication. We control for the index medication and patient clinical factors (described below) in an effort to strike a balance between our chief interest in general practice patterns and the nuanced complexity of psychiatric deprescribing.

*Covariates.* This study used the same extensive list of covariates as used in the earlier studies of this cohort [20,23], which were selected based on their potential to impact the type of prescriber seen by the patients and their role in the Andersen Behavioral Model of Healthcare Utilization [27,28]. Predisposing factors included age at initiation date and patient sex as identified on their insurance. Enabling factors included employment status, employment type, and the relationship of the insured individual to the employee. Need factors included a mix of mental health conditions, physical health conditions associated with prescribing complexity (e.g., liver disease), and the Charlson Comorbidity Index (CCI) [29]. Healthcare utilization factors assessed prior to treatment initiation included the number of psychotherapy visits, psychiatric emergency department visits, and the index medication. Additional contextual factors included rurality (based on metropolitan statistical areas), state of residence, and the year in which treatment began. A detailed description of these covariates is available elsewhere [23].

Note that all covariates were measured in the 6 months prior to treatment initiation. Given that the goal of the study was to assess the impact of prescriber type on deprescribing, covariates were only included if they had the potential to be causally associated with both the provider type and deprescribing based on their temporal ordering (e.g., adherence occurs after the provider type has been determined and therefore is not capable of being causally associated with the provider type). Future studies may wish to examine the potential for health events during treatment initiation to mediate or moderate the association between provider type and deprescribing; however, such an assessment falls outside the scope of the present study. As such, we did not adjust for any conditions or health events occurring during the 90-day initiation period since the provider type selection had already occurred.

## Analyses

Two comparison cohorts were created using inverse for propensity for treatment weighting (IPTW) to compare prescribing psychologists to psychiatrists (the psychiatrist cohort) and prescribing psychologists to PCPs (the PCP cohort; Fig 2). When paired with a strong study design, the use of IPTW weights has been shown to produce estimates similar to those that would be obtained through a randomized controlled trial [30,31].

Stabilized IPTW weights were created by first calculating the probability of having a given prescriber type based on the full set of covariates and then limiting both cohorts to the area of common support (i.e., the range of propensity scores with a non-zero probability of treatment for both treated and untreated patients). Then, we asymmetrically trimmed the 99th percentile of prescribing psychologist patients and the 1st percentile of psychiatrist and PCP patients for each cohort to reduce the impact of unmeasured confounding [31]. Finally, the IPTW weights were then re-estimated in the trimmed cohorts and checked for covariate balance using standardized mean differences for continuous and binary variables and

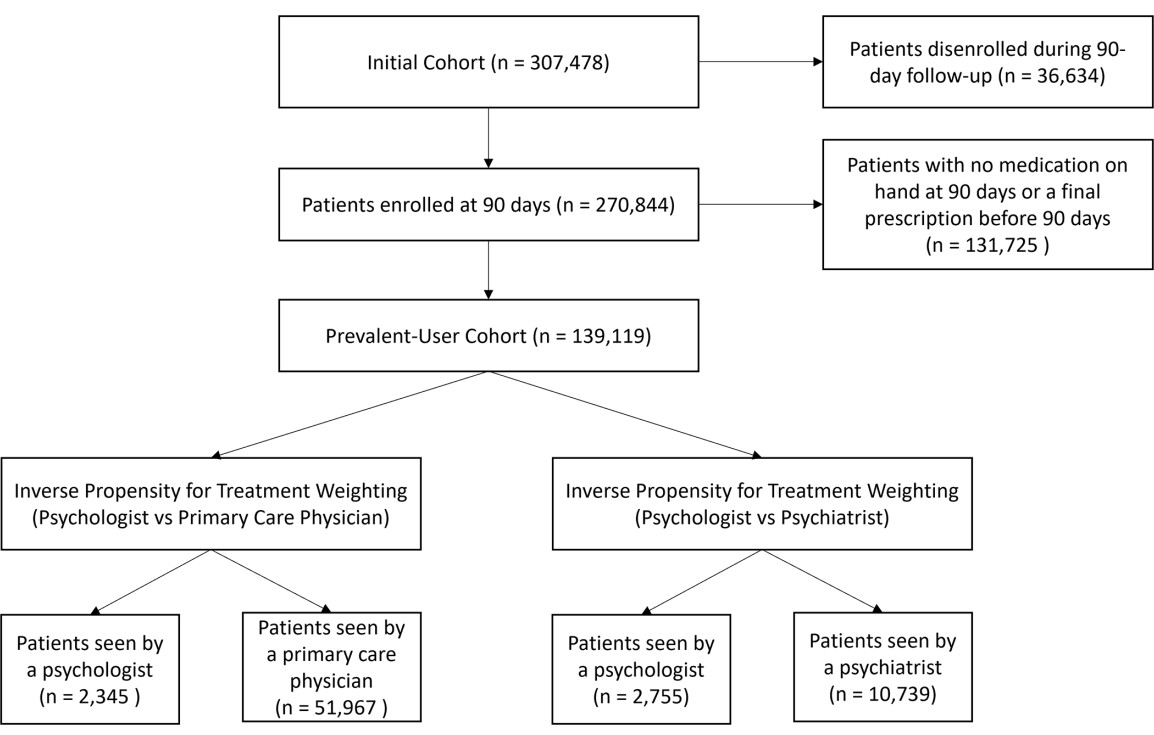

**Fig 2. Inclusion and exclusion criteria.**

a qualitative assessment of categorical variables. We also searched for outliers with large IPTW weights (weight > 10) and found none [31].

Analysis was conducted in SAS Studio v.3.8. First, we weighted the cohorts and estimated Kaplan-Meier (KM) curves, log-rank tests for differences in the curves, and mean time until deprescribing outcome by provider type. Second, we estimated Cox proportional hazards models to compare rates of deprescribing outcomes between provider types. We used a "doubly-robust" approach for these models [32], where IPTW weights and traditional covariate adjustment were used conjointly to increase protection against bias due to model misspecification. Additionally, any covariates that were rare (<1%) among patients of prescribing psychologists (the smallest prescriber group) were excluded from the models to protect against unstable estimates. Missing data for categorical variables were coded as a distinct category; no data for continuous variables were missing. The proportional hazards assumption of the Cox model was supported by a visual assessment of the KM curves.

## Results

Among patients in the psychiatrist cohort, 2,755 saw a prescribing psychologist and 10,793 saw a psychiatrist. Of patients in the PCP cohort, 2,345 saw a prescribing psychologist and 103,933 saw a PCP. Pre-weighting and post-weighting differences in baseline characteristics can be found in the supplemental materials S1 Text (Psychiatrists: Tables A-G, Fig A; PCPs: Table A, Tables H-M, Fig B). The IPTW weighting produced good covariate balance with only minor differences in categorical variables and no indication of outliers, suggesting that the IPTW weights successfully addressed baseline confounding for the available covariates.

### Deprescribing without replacement

In the psychiatrist cohort, deprescribing as measured in gaps of 30 days was common among patients of both prescribing psychologists (52.9%) and psychiatrists (47.6%), with mean times to deprescribing of 254.2 days (SE = 2.98) and 263.9 days (SE = 1.20), respectively. In the weighted KM curve, the rate of gaps was higher for patients of psychologists ($p$ = .0017; Fig A in S2 Text). This was consistent in the doubly-robust model (Table 1), with the rate of gaps being 13% higher among patients of prescribing psychologists (Hazard Ratio (HR) = 1.13, 95% CI = [1.06, 1.20]). The direction of this relationship was maintained but not statistically significant in the sensitivity analyses requiring 90-day and 180-day gaps (90-day: 1.06 [0.99, 1.13]; 180-day: 1.04 [0.95, 1.13]). The result was consistent, however, in both sensitivity analyses regarding the identification of provider type (strict hierarchy: 1.11 [1.04, 1.18]; recent provider: 1.09 [1.03, 1.16]).

Gaps of at least 30 days were common in the PCP cohort for patients of prescribing psychologists (52.3%) and even more so for patients of PCPs (64.0%), with mean times to deprescribing of 251.2 days (SE = 4.33) and 220.8 days (SE = 0.40), respectively. The weighted KM curve demonstrated a significantly lower rate of gaps for prescribing psychologists ($p$ < .0001; Fig B in S2 Text). In the doubly-robust model (Table 2), prescribing psychologists had a 27% lower rate of gaps in prescribing (0.73 [0.69, 0.78]). This finding was highly consistent in sensitivity analyses, producing similar results in the 90-day (0.75 [0.70, 0.80]), 180-day (0.78 [0.72, 0.85]), strict hierarchy (0.69 [0.64, 0.74]), and recent provider (0.71, [0.67, 0.76]) sensitivity analyses.

### Complete discontinuation of prescribing

Approximately half of patients in the psychiatrist cohort discontinued prescribing within a year (prescribing psychologists: 50.1%; psychiatrists: 50.0%). Patients of prescribing psychologists had a mean of 267.6 days (SE = 2.93) to discontinuation and patients of psychiatrists had a mean of 264.0 days (SE = 1.21), and the difference in the rate of discontinuation was not significant ($p$ = 0.1412; Fig C in S2 Text). This was supported further by a non-significant hazard ratio (0.97 [0.92, 1.03]; Table 1) that was also consistent for both alternative provider identification methodologies in the sensitivity analysis (strict hierarchy: 0.98 [0.91, 1.04]; recent provider: 0.95 [0.89, 1.01].

**Table 1. Hazard ratios and 95% confidence intervals comparing deprescribing rates between prescribing psychologists and psychiatrists.**

| | Medication Gap (30 Days) | | | Any Reduction in Days' Supply | | | Discontinuation of Prescribing | | |
|---|---|---|---|---|---|---|---|---|---|
| | | 95% Confidence Interval | | | 95% Confidence Interval | | | 95% Confidence Interval | |
| | HR | Lower | Upper | HR | Lower | Upper | HR | Lower | Upper |
| **Provider Type** | | | | | | | | | |
| Prescribing Psychologist | 1.13 | 1.06 | 1.20 | 0.79 | 0.69 | 0.91 | 0.97 | 0.92 | 1.03 |
| **Predisposing** | | | | | | | | | |
| Age at Service Date* | 0.99 | 0.98 | 0.99 | 1.01 | 1.00 | 1.01 | 0.99 | 0.99 | 0.99 |
| Male | 1.08 | 1.03 | 1.14 | 0.75 | 0.67 | 0.84 | 1.06 | 1.01 | 1.11 |
| **Enabling** | | | | | | | | | |
| Employment Status | | | | | | | | | |
| Part-Time | 0.77 | 0.55 | 1.08 | 0.92 | 0.45 | 1.88 | 0.81 | 0.55 | 1.20 |
| Retired | 0.89 | 0.69 | 1.16 | 0.93 | 0.58 | 1.49 | 0.81 | 0.59 | 1.10 |
| Other/Unknown | 0.92 | 0.85 | 1.00 | 0.96 | 0.81 | 1.14 | 1.35 | 1.24 | 1.47 |
| Employment Type | | | | | | | | | |
| Hourly | 1.11 | 1.00 | 1.22 | 1.04 | 0.83 | 1.31 | 1.01 | 0.90 | 1.13 |
| Unknown | 1.12 | 1.01 | 1.24 | 1.13 | 0.91 | 1.40 | 1.16 | 1.04 | 1.30 |
| Relation to Employee | | | | | | | | | |
| Spouse | 0.90 | 0.83 | 0.97 | 1.08 | 0.94 | 1.25 | 0.86 | 0.80 | 0.92 |
| Child/Other Dependent | 0.98 | 0.90 | 1.06 | 1.12 | 0.93 | 1.35 | 0.74 | 0.68 | 0.80 |
| **Need** | | | | | | | | | |
| Mental Health Diagnoses | | | | | | | | | |
| Schizophrenia/Psychotic Disorders | -- | -- | -- | -- | -- | -- | -- | -- | -- |
| Bipolar Disorders | 0.80 | 0.70 | 0.91 | 1.10 | 0.89 | 1.35 | 1.01 | 0.91 | 1.13 |
| Depressive Disorders | 1.12 | 1.06 | 1.20 | 0.97 | 0.85 | 1.11 | 1.10 | 1.04 | 1.17 |
| Anxiety Disorders | 1.07 | 1.01 | 1.14 | 0.98 | 0.86 | 1.13 | 1.04 | 0.98 | 1.11 |
| Post-Traumatic Stress Disorder | 0.99 | 0.84 | 1.15 | 0.74 | 0.53 | 1.04 | 1.02 | 0.88 | 1.18 |
| Eating Disorders | -- | -- | -- | -- | -- | -- | -- | -- | -- |
| Personality Disorders | 0.98 | 0.86 | 1.11 | 0.87 | 0.66 | 1.14 | 0.98 | 0.87 | 1.11 |
| Autism Spectrum Disorder | 0.70 | 0.55 | 0.88 | 1.54 | 0.99 | 2.40 | 0.85 | 0.68 | 1.07 |
| ADHD | 0.94 | 0.87 | 1.01 | 0.82 | 0.68 | 1.00 | 0.94 | 0.87 | 1.01 |
| Conduct Disorder | 0.88 | 0.75 | 1.03 | 1.04 | 0.70 | 1.55 | 0.94 | 0.79 | 1.12 |
| Physical Diagnoses | | | | | | | | | |
| Epilepsy | -- | -- | -- | -- | -- | -- | -- | -- | -- |
| Hypertension | 1.12 | 0.98 | 1.27 | 0.87 | 0.66 | 1.14 | 1.04 | 0.92 | 1.17 |
| Congestive Heart Failure | -- | -- | -- | -- | -- | -- | -- | -- | -- |
| Liver Disease | 1.12 | 0.98 | 1.28 | 1.01 | 0.73 | 1.38 | 0.86 | 0.74 | 1.00 |
| Diabetes | 1.13 | 0.95 | 1.35 | 1.05 | 0.74 | 1.49 | 0.95 | 0.80 | 1.13 |
| Cancer | -- | -- | -- | -- | -- | -- | -- | -- | -- |
| Insomnia | -- | -- | -- | -- | -- | -- | -- | -- | -- |
| Charlson Comorbidity Index | 1.08 | 0.91 | 1.29 | 0.77 | 0.50 | 1.18 | 1.29 | 1.10 | 1.51 |
| **Context** | | | | | | | | | |
| Rurality | | | | | | | | | |
| Metropolitan Statistical Area | 0.92 | 0.85 | 1.00 | 1.28 | 1.06 | 1.55 | 0.90 | 0.83 | 0.98 |
| Unknown | 1.63 | 1.35 | 1.96 | 1.19 | 0.66 | 2.16 | 2.14 | 1.82 | 2.51 |
| State of Residence - New Mexico | 1.03 | 0.98 | 1.10 | 1.39 | 1.24 | 1.57 | 1.09 | 1.03 | 1.16 |
| Year | 1.00 | 0.99 | 1.01 | 1.03 | 1.01 | 1.05 | 1.08 | 1.07 | 1.09 |

*(Continued)*

PLOS Mental Health

**Table 1.** (Continued)

| | Medication Gap (30 Days) | | | Any Reduction in Days' Supply | | | Discontinuation of Prescribing | | |
|---|---|---|---|---|---|---|---|---|---|
| | | 95% Confidence Interval | | | 95% Confidence Interval | | | 95% Confidence Interval | |
| | HR | Lower | Upper | HR | Lower | Upper | HR | Lower | Upper |
| **Healthcare Utilization** | | | | | | | | | |
| Psychotherapy | | | | | | | | | |
| 1-4 Visits | 0.86 | 0.81 | 0.91 | 1.09 | 0.96 | 1.24 | 0.86 | 0.81 | 0.91 |
| 5+Visits | 0.96 | 0.89 | 1.05 | 0.94 | 0.77 | 1.14 | 0.98 | 0.90 | 1.06 |
| Any Psychiatric ED visits | -- | -- | -- | -- | -- | -- | -- | -- | -- |
| **Psychotropic Medications** | | | | | | | | | |
| Antidepressants | 1.20 | 1.10 | 1.31 | 0.63 | 0.54 | 0.73 | 0.93 | 0.86 | 1.01 |
| Antipsychotics/Tranquilizers | 0.85 | 0.74 | 0.98 | 0.90 | 0.71 | 1.15 | 0.88 | 0.78 | 1.00 |
| Anxiolytics/Sedatives/Hypnotics | 1.02 | 0.91 | 1.14 | 1.00 | 0.83 | 1.19 | 0.92 | 0.83 | 1.01 |
| Hypotensive Agents | 0.90 | 0.74 | 1.09 | 0.90 | 0.62 | 1.33 | 0.78 | 0.64 | 0.94 |
| Stimulants | 1.33 | 1.21 | 1.46 | 0.46 | 0.37 | 0.57 | 0.85 | 0.78 | 0.93 |

As in the psychiatry cohort, prescribing had been discontinued for approximately half of the PCP cohort within the year (prescribing psychologists: 49.0%; PCPs: 51.3%). The mean time to discontinuation for patients of prescribing psychologists was 262.6 days (SE=4.28) and 256.8 days (SE=0.40) for patients of PCPs. The weighted KM curve demonstrated no significant difference in the rate of discontinuation ($p=0.4842$; Fig D in S2 Text), and this was further supported by the adjusted model (0.96 [0.90, 1.02]; Table 2). However, the rate of discontinuation was lower for patients of prescribing psychologists in both the strict hierarchy (0.88 [0.83, 0.96]) and recent provider (0.92 [0.87, 0.98]) sensitivity analyses.

### Sustained reduction in days' supply

There was a modest number of sustained reductions in the psychiatrist cohort, being slightly less common among patients of prescribing psychologists (7.8%) than psychiatrists (10.6%). The mean time to deintensification was 342.0 days (SE=1.86) among patients of prescribing psychologists and 337.8 days (SE=0.83) among patients of psychiatrists, and the difference in the rate of sustained reductions was significant (p=0.0123; Fig E in S2 Text). The doubly-robust model supported this (Table 1), with prescribing psychologists having a 21% lower rate of sustained reductions (0.79 [0.69, 0.91]). This result was consistent in the sensitivity analysis requiring a supply reduction of at least 7 days (0.81 [0.70, 0.94]). In the sensitivity analysis for provider identification method, the association was no longer significant when the stringent hierarchy approach was used (0.92 [0.80, 1.06]), but was consistent in the recent provider approach (0.80 [0.70, 0.92]).

In the PCP cohort, sustained reductions were relatively uncommon for patients of prescribing psychologists (8.0%) and PCPs (5.5%), with mean times to deintensification of 341.6 days (SE=2.71) and 350.8 days (SE=0.20), respectively. This difference in the rate of sustained reductions was significant (p<.0001; Fig F in S2 Text). The model supported this (Table 2), with patients of prescribing psychologists experiencing sustained reductions at a rate 64% higher than those of PCPs (1.64 [1.42, 1.90]). Furthermore, this finding was consistent in the sensitivity analysis requiring a 7-day reduction in supply (1.62 [1.38, 1.89]), as well as the strict hierarchy (1.62 [1.37, 1.92]) and recent provider (1.65 [1.42, 1.92]) sensitivity analyses.

## Discussion

There is a growing interest in deprescribing psychotropic medications within the mental health community, and advocates for RxP have argued that deprescribing is a task for which prescribing psychologists are well positioned to lead. This study

 

**Table 2. Hazard ratios and 95% confidence intervals comparing deprescribing rates between prescribing psychologists and primary care physicians.**

| | Medication Gap (30 Days) | | | Any Reduction in Days' Supply | | | Discontinuation of Prescribing | | |
|---|---|---|---|---|---|---|---|---|---|
| | | 95% Confidence Interval | | | 95% Confidence Interval | | | 95% Confidence Interval | |
| | HR | Lower | Upper | HR | Lower | Upper | HR | Lower | Upper |
| **Provider Type** | | | | | | | | | |
| Prescribing Psychologist | 0.73 | 0.69 | 0.78 | 1.64 | 1.42 | 1.90 | 0.96 | 0.90 | 1.02 |
| **Predisposing** | | | | | | | | | |
| Age at Service Date* | 0.99 | 0.99 | 0.99 | 1.00 | 1.00 | 1.01 | 0.99 | 0.99 | 0.99 |
| Male | 1.05 | 1.04 | 1.07 | 0.95 | 0.90 | 1.00 | 1.06 | 1.04 | 1.08 |
| **Enabling** | | | | | | | | | |
| Employment Status | | | | | | | | | |
| Part-Time | 1.06 | 0.95 | 1.19 | 1.13 | 0.80 | 1.59 | 1.31 | 1.15 | 1.50 |
| Retired | 0.94 | 0.86 | 1.02 | 1.15 | 0.92 | 1.44 | 1.03 | 0.93 | 1.15 |
| Other/Unknown | 1.05 | 1.03 | 1.08 | 0.88 | 0.81 | 0.96 | 1.46 | 1.42 | 1.51 |
| Employment Type | | | | | | | | | |
| Hourly | 1.07 | 1.03 | 1.10 | 0.97 | 0.88 | 1.08 | 1.01 | 0.97 | 1.05 |
| Unknown | 1.04 | 1.00 | 1.07 | 1.00 | 0.90 | 1.11 | 1.19 | 1.14 | 1.24 |
| Relation to Employee | | | | | | | | | |
| Spouse | 0.92 | 0.90 | 0.94 | 1.10 | 1.03 | 1.17 | 0.89 | 0.87 | 0.91 |
| Child/Other Dependent | 1.06 | 1.03 | 1.09 | 1.04 | 0.94 | 1.15 | 0.84 | 0.81 | 0.87 |
| **Need** | | | | | | | | | |
| Mental Health Diagnoses | | | | | | | | | |
| Schizophrenia/Psychotic Disorders | -- | -- | -- | -- | -- | -- | -- | -- | -- |
| Bipolar Disorders | 0.61 | 0.54 | 0.70 | 1.75 | 1.38 | 2.22 | 1.04 | 0.93 | 1.16 |
| Depressive Disorders | 1.00 | 0.97 | 1.03 | 1.04 | 0.94 | 1.15 | 1.06 | 1.02 | 1.10 |
| Anxiety Disorders | 0.98 | 0.95 | 1.01 | 1.08 | 0.99 | 1.18 | 1.05 | 1.02 | 1.08 |
| Post-Traumatic Stress Disorder | 1.05 | 0.91 | 1.20 | 1.18 | 0.84 | 1.68 | 1.08 | 0.93 | 1.24 |
| Eating Disorders | -- | -- | -- | -- | -- | -- | -- | -- | -- |
| Personality Disorders | 0.84 | 0.77 | 0.92 | 1.03 | 0.80 | 1.33 | 0.93 | 0.85 | 1.02 |
| Autism Spectrum Disorder | 1 | 0.949 | 1.013 | 0.9 | 0.789 | 1.045 | 0.9 | 0.885 | 0.956 |
| ADHD | 0.59 | 0.49 | 0.71 | 2.08 | 1.40 | 3.11 | 0.77 | 0.63 | 0.94 |
| Conduct Disorder | 0.71 | 0.63 | 0.80 | 1.38 | 0.97 | 1.95 | 0.86 | 0.75 | 0.99 |
| Physical Diagnoses | | | | | | | | | |
| Epilepsy | -- | -- | -- | -- | -- | -- | -- | -- | -- |
| Hypertension | 1.15 | 1.11 | 1.19 | 1.02 | 0.91 | 1.15 | 1.07 | 1.03 | 1.11 |
| Congestive Heart Failure | -- | -- | -- | -- | -- | -- | -- | -- | -- |
| Liver Disease | 1.05 | 1.01 | 1.10 | 0.90 | 0.76 | 1.05 | 0.87 | 0.82 | 0.92 |
| Diabetes | 1.01 | 0.96 | 1.06 | 1.07 | 0.92 | 1.24 | 0.97 | 0.92 | 1.02 |
| Cancer | 1.04 | 0.97 | 1.12 | 1.10 | 0.89 | 1.36 | 1.11 | 1.03 | 1.20 |
| Insomnia | -- | -- | -- | -- | -- | -- | -- | -- | -- |
| Charlson Comorbidity Index | 1.01 | 0.96 | 1.07 | 1.14 | 0.97 | 1.35 | 1.10 | 1.04 | 1.17 |
| **Context** | | | | | | | | | |
| Rurality | | | | | | | | | |
| Metropolitan Statistical Area | 0.97 | 0.95 | 0.99 | 1.13 | 1.05 | 1.22 | 0.97 | 0.95 | 0.99 |
| Unknown | 1.43 | 1.36 | 1.51 | 1.41 | 1.12 | 1.78 | 2.11 | 2.00 | 2.22 |

*(Continued)*

**Table 2.** (Continued)

| | Medication Gap (30 Days) | | | Any Reduction in Days' Supply | | | Discontinuation of Prescribing | | |
|---|---|---|---|---|---|---|---|---|---|
| | | 95% Confidence Interval | | | 95% Confidence Interval | | | 95% Confidence Interval | |
| | HR | Lower | Upper | HR | Lower | Upper | HR | Lower | Upper |
| State of Residence - New Mexico | 1.11 | 1.09 | 1.13 | 1.36 | 1.28 | 1.44 | 1.24 | 1.21 | 1.27 |
| Year | 1.01 | 1.01 | 1.01 | 1.03 | 1.02 | 1.04 | 1.10 | 1.10 | 1.11 |
| **Healthcare Utilization** | | | | | | | | | |
| Psychotherapy | | | | | | | | | |
| 1-4 Visits | 0.86 | 0.82 | 0.90 | 1.15 | 1.01 | 1.32 | 0.90 | 0.86 | 0.95 |
| 5+ Visits | 0.86 | 0.81 | 0.92 | 1.20 | 1.00 | 1.44 | 0.86 | 0.80 | 0.92 |
| Any Psychiatric ED visits | -- | -- | -- | -- | -- | -- | -- | -- | -- |
| **Psychotropic Medications** | | | | | | | | | |
| Antidepressants | 1.02 | 1.00 | 1.05 | 0.67 | 0.63 | 0.73 | 0.94 | 0.92 | 0.97 |
| Antipsychotics/Tranquilizers | 0.68 | 0.63 | 0.75 | 1.46 | 1.21 | 1.76 | 0.92 | 0.85 | 1.01 |
| Anxiolytics/Sedatives/Hypnotics | 1.07 | 1.04 | 1.10 | 0.99 | 0.91 | 1.08 | 0.93 | 0.90 | 0.96 |
| Hypotensive Agents | 0.81 | 0.75 | 0.86 | 0.84 | 0.69 | 1.03 | 0.82 | 0.76 | 0.88 |
| Stimulants | 1.01 | 0.98 | 1.04 | 0.42 | 0.38 | 0.47 | 0.77 | 0.74 | 0.80 |

described differences in the rates of deprescribing between prescribing psychologists, psychiatrists, and PCPs under the hypothesis that prescribing psychologists would have the highest rate of deprescribing. Contrary to our hypothesis, our findings suggest that prescribing psychologists deprescribe at a rate in between psychiatrists and PCPs for deprescribing without replacement and deintensification, but complete discontinuation at similar rates for all providers. Furthermore, the prescriber type with the highest rates of deprescribing depends upon the deprescribing outcome being examined. When deprescribing is defined based on medication gaps, PCPs have the highest rates of deprescribing followed by prescribing psychologists and psychiatrists. When deprescribing is examined through a deintensification lens and defined as sustained reductions in days' supply, psychiatrists have the highest rate of deprescribing, followed by prescribing psychologists and PCPs. These findings have several implications for mental health practice, policy, and research.

Broadly, deprescribing as identified by either a gap in prescription claims or by a complete discontinuation of prescribing appears to be occurring with regularity among patients receiving psychotropic medications. While some patients may have conditions needing long-term management with psychotropic medications, the high degree of deprescribing occurring within the first year (after a 90-day initiation period) of starting a psychotropic medication may reflect the growing emphasis on deprescribing among providers [4–6]. However, this may also reflect the adverse physical and psychological effects that contribute to many patients' desire to discontinue these medications [33]. Given that known barriers regarding technical aspects of deprescribing are seemingly more likely to impact PCPs [5,12], the finding that PCPs were the most common deprescribers when defined based on medication gaps was unexpected, and could actually be a signal of greater nonadherence among their patients as seen previously [20]. Conversely, deprescribing in the form of deintensification was less common overall, but more common among psychiatrists and prescribing psychologists than PCPs. Combined, these findings may suggest that deprescribing in the form of medication gaps is a less technically challenging approach to deprescribing (e.g., taking breaks from medication to see how the patient fares), whereas systematic deintensification through reduced days' supply may reflect a more technically nuanced approach to deprescribing (e.g., tapering via switching to a short course of lower dose medications or via pill splitting) preferred by mental health specialists [34], Future research is needed to understand which approaches providers take to psychotropic deprescribing, why, and how patients fare under those different approaches.

Contrary to expectation, prescribing psychologists did not have the highest rates of deprescribing, falling in between psychiatrists and PCPs on deprescribing without replacement and supply reduction while having a similar rate of complete discontinuation. However, while prescribing psychologists may not be deprescribing more than all providers, their consistent placement between psychiatrists and PCPs offers promise in terms of the goal of RxP to increase access to high-quality mental health prescribing. Prior research has documented the potential for RxP to reduce prescriber shortages [35], and the present findings suggest that this increased access would include deprescribing that is at least comparable to other prescribers. As such, while prescribing psychologists do not appear to have reset the bar for deprescribing, their deprescribing practices are at least firmly within the range of other prescribers. Policymakers considering RxP should take this comparability into account when considering the potential impact of adopting such a policy.

Furthermore, given that prior research has demonstrated that prescribing psychologists are viewed as safe prescribers by their colleagues [36,37] while having lower rates of adverse drug events and psychotropic polypharmacy than psychiatrists [20], it seems unlikely that different rates of deprescribing among prescribing psychologists are due to technical issues of deprescribing. Additional research is needed to elucidate the mechanisms behind these deprescribing differences. Based on prior findings, we posit that the lower rate of deprescribing may reflect a more cautious approach to medication initiation used by prescribing psychologists as well as a tendency to pair medication with psychotherapy. We hypothesize that lower rates of psychotropic polypharmacy among patients of prescribing psychologists inherently reduces the number of medications from which an individual can be deprescribed, resulting in fewer opportunities to deprescribe rather than a lower proclivity for deprescribing. Alternatively, while prescribing psychologists are safe prescribers, this may indicate that deprescribing remains an area of expertise in which psychiatrists remain uniquely skilled. Further complicating the interpretation of this relationship is that our use of insurance claims precludes us from assessing the reason for deprescribing. It is possible that different provider types are more adept at identifying when a medication is no longer needed, while others may be more attuned to patient feedback regarding symptoms, and those different rationales for deprescribing may lead to different deprescribing approaches. Differences in deprescribing may also reflect resource variation within their respective clinical settings. For example, PCPs practicing in interdisciplinary primary care practices may have access to clinical pharmacists who can highlight deprescribing opportunities, while psychiatrists and prescribing psychologists in private practice may not. Qualitative interviews with prescribing psychologists, psychiatrists, and PCPs regarding their deprescribing experiences may shed light on the unique relationships between each provider type and deprescribing.

Finally, the finding that which providers deprescribed most varied substantially by how deprescribing was defined suggests a need for continued research into defining and identifying deprescribing. Prior research has demonstrated the potential for developing composite measures of deprescribing for specific drug classes (e.g., antidiabetic therapies) in the nursing home population [38], and similar efforts are needed in the context of psychotropic medications and mental health. For example, a study linking claims data and electronic health records may provide an opportunity to develop and validate an algorithm for identifying different approaches to deprescribing psychotropic medications. Definitions such as those used in this study could be applied to identify potential cases of deprescribing in claims, while a chart note review by providers could identify true deprescribing cases to serve as a "gold standard" by which to estimate the validity and reliability of the algorithm. Additionally, given the chronicity of some medications used in mental health, a Delphi study in which mental health service providers reach a consensus definition of deprescribing in the mental health context may demonstrate that deprescribing should be examined differently in this population. Similar research is emerging for identifying low-value prescribing in older adults as an extension of the deprescribing literature in that population [39], and a corresponding effort in mental health would provide a similar benefit to research in this area.

## Limitations

This study has limitations worth consideration. First, we were unable to measure if patients change providers within provider types (e.g., from one PCP to another) or between provider types (e.g., a PCP referring to a Psychiatrist). There

are almost certainly patients who will seek care from different providers in both ways, such as an individual who received their medical care from an urgent care center and has a different provider for most encounters (within-provider type) or an individual who was initially treated by a PCP while waiting for a psychiatrist to become available (between-provider types). For these individuals, this provider switching may impact the rate of deprescribing depending on the preferences of the individual providers. Furthermore, the often-fragmented nature of mental health care can result in scenarios where a patient may be receiving different treatments from different providers that accumulate over time in a way that does not reflect the beliefs or expectations of the provider currently managing their care. For example, initial prescriptions by a PCP may continue to be refilled even though the patient is now also in the care of a mental health specialist. We were unable to assess provider switching and the subsequent complexities given that individual prescribers are not linked to claims in the MarketScan data. The algorithm-based approach used to identify the initial prescriber is limited to identifying the prescriber type rather than the individual prescriber directly, and we would therefore be unable to assess provider switching with this approach (either between or within provider types). This would be further complicated by the potential for medications to have multiple refill orders. It is probable that there were patients in this study who did not see their prescriber again after the 90-day initiation window but still saw other providers during that time for their other health needs.

Second, studying deprescribing is still a relatively nascent field and measuring deprescribing in claims remains a considerable methodological challenge [26]. For example, patients who switched between multiple drug classes were not identified as deprescribed, as they were receiving a potential replacement psychotropic medication; however, the clinical reality of this process is much more complex than can be captured through claims. Claims data precludes differentiation between a medication that was deprescribed because it was intended to be a short-term solution versus a medication that was intended to be a long-term solution but was deprescribed because it was ineffective. Similarly, a patient may decide to discontinue medication without consulting their provider or against medical advice, which would still appear to be deprescribing as viewed in claims.

Third, there are aspects of our cohort selection that limit generalization. Given our use of private, employer-sponsored insurance claims, our results may not generalize well to publicly-insured patients covered by Medicaid or Medicare. This limitation would both reduce the representation of low-income and senior participants in the sample. In particular, a notably larger proportion of adults with mental illness are insured through Medicaid (33%) than private insurance (21%) [40], suggesting a need for future research to replicate this study using Medicaid claims. Similarly, the present study did not include advanced practice providers. Given the rapid growth of the psychiatric mental health nurse practitioner workforce [41], there is a need for future research to examine the deprescribing practices of those providers relative to psychiatrists, PCPs, non-psychiatric nurse practitioners, and physician associates.

Finally, the use of a look-back period to establish prevalence prevents the inclusion of deprescribing that occurred during that time. The initial period of psychotropic medication use is likely a volatile time for medication changes, making it plausible that some deprescribing is missed if patients begin medication and quickly determine that psychotherapy may be a better solution.

## Conclusion

The rate of psychotropic deprescribing varies by prescriber type and depends significantly on how deprescribing is defined. Prescribing psychologists were consistently deprescribing at a rate between that of PCPs and psychiatrists, possibly resulting from a difference in their approach to medication use. Considerable future research is needed to identify how deprescribing should be approached in the mental health context and why deprescribing rates vary by prescriber type.

## Supporting information

**S1 Text. Propensity Score Covariate Balance.** Table A: Covariate Balance Before Weighting. Table B: Provider Type by Year (Psychiatrist Cohort). Table C. Provider Type by Employment Status (Psychiatrist Cohort). Table D: Provider Type by Salary Type (Psychiatrist Cohort).Table E: Provider Type by Relationship to Employee (Psychiatrist Cohort).

Table F: Provider Type by Rurality (Psychiatrist Cohort). Table G: Provider Type by Index Medication Type (Psychiatrist Cohort). Table H: Provider Type by Year (Primary Care Physician Cohort). Table I. Provider Type by Employment Status (Primary Care Physician Cohort). Table J: Provider Type by Salary Type (Primary Care Physician Cohort). Table K: Provider Type by Relationship to Employee (Primary Care Physician Cohort). Table L: Provider Type by Rurality (Primary Care Physician Cohort). Table M: Provider Type by Index Medication Type (Primary Care Physician Cohort). Fig A: Covariate Balance After Weighting (Psychiatrist Cohort). Fig B: Covariate Balance After Weighting (Primary Care Physician Cohort).
(DOCX)

**S2 Text. Additional Results.** Fig A. Days Discontinuation Without Replacement Among Patients of Prescribing Psychologists and Psychiatrists. Fig B. Days Discontinuation Without Replacement Among Patients of Prescribing Psychologists and Primary Care Physicians. Fig C. Days Until Complete Discontinuation of Prescribing Among Patients of Prescribing Psychologists and Psychiatrists. Fig D. Days Until Complete Discontinuation of Prescribing Among Patients of Prescribing Psychologists and Primary Care Physicians. Fig E. Days Until Sustained Reduction in Days' Supply Among Patients of Prescribing Psychologists and Psychiatrists. Fig F. Days Until Sustained Reduction in Days' Supply Among Patients of Prescribing Psychologists and Primary Care Physicians.
(DOCX)

## Author contributions

**Conceptualization:** Phillip M. Hughes, Kathleen C. Thomas.

**Formal analysis:** Phillip M. Hughes.

**Investigation:** Robert E. McGrath, Robert B. Christian, Kathleen C. Thomas.

**Methodology:** Phillip M. Hughes, Joshua D. Niznik, Casey R. Tak, Kathleen C. Thomas.

**Supervision:** Joshua D. Niznik, Robert E. McGrath, Casey R. Tak, Robert B. Christian, Betsy L. Sleath, Kathleen C. Thomas.

**Writing – original draft:** Phillip M. Hughes.

**Writing – review & editing:** Phillip M. Hughes, Joshua D. Niznik, Robert E. McGrath, Casey R. Tak, Robert B. Christian, Betsy L. Sleath, Kathleen C. Thomas.

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
