## [Decision Letter · Decision Letter 0]

21 Oct 2025

PMEN-D-25-00326

Psychotropic deprescribing across different prescribing professions in New Mexico and Louisiana

PLOS Mental Health

Dear Dr. Hughes,

Thank you for submitting your manuscript to PLOS Mental Health and I am very sorry for the delay. Thank you for your patience.  After careful consideration of the reviewer reports, which we have now received, we feel that your paper has merit but does not fully meet PLOS Mental Health’s publication criteria as it currently stands. Therefore, we invite you to submit a revised version of the manuscript that addresses the points raised during the review process.

Please address all of the points raised, which you can find below.

We look forward to receiving your revised manuscript.

Kind regards,

Dr Karli Montague-Cardoso

Executive Editor

PLOS Mental Health

Journal Requirements:

1. Please note that PLOS Mental Health has specific guidelines on code sharing for submissions in which author-generated code underpins the findings in the manuscript. In these cases, we expect all author-generated code to be made available without restrictions upon publication of the work. Please review our guidelines at https://journals.plos.org/mentalhealth/s/materials-and-software-sharing#loc-sharing-code and ensure that your code is shared in a way that follows best practice and facilitates reproducibility and reuse.

2. Please send a completed 'Competing Interests' statement, including any COIs declared by your co-authors. If you have no competing interests to declare, please state "The authors have declared that no competing interests exist". Otherwise please declare all competing interests beginning with the statement "I have read the journal's policy and the authors of this manuscript have the following competing interests:"

3. Please amend your detailed Financial Disclosure statement. This is published with the article. It must therefore be completed in full sentences and contain the exact wording you wish to be published.

1. Please clarify all sources of funding (financial or material support) for your study. List the grants (with grant number) or organizations (with url) that supported your study, including funding received from your institution. 

2. State the initials, alongside each funding source, of each author to receive each grant.

3. State what role the funders took in the study. If the funders had no role in your study, please state: “The funders had no role in study design, data collection and analysis, decision to publish, or preparation of the manuscript.”

4. If any authors received a salary from any of your funders, please state which authors and which funders.

4. We do not publish any copyright or trademark symbols that usually accompany proprietary names, eg (R), (C), or TM  (e.g. next to drug or reagent names). Please remove all instances of trademark/copyright symbols throughout the text, including ® on page 6, 9.

5. We have noticed that you have cited Table 10, 11 in the manuscript file but there are no corresponding tables in the manuscript. Please amend your manuscript to include this table, noting that tables should not be uploaded as individual files.

Reviewers' comments:

Reviewer's Responses to Questions

**Comments to the Author**

1. Does this manuscript meet PLOS Mental Health’s publication criteria?

Reviewer #1: Yes

Reviewer #2: Partly

2. Has the statistical analysis been performed appropriately and rigorously?

Reviewer #1: Yes

Reviewer #2: I don't know

3. Have the authors made all data underlying the findings in their manuscript fully available (please refer to the Data Availability Statement at the start of the manuscript PDF file)?

Reviewer #1: Yes

Reviewer #2: Yes

4. Is the manuscript presented in an intelligible fashion and written in standard English?

Reviewer #1: No

Reviewer #2: Yes

Reviewer #1: The manuscript utilizes a large database to investigate deprescribing patterns among three prescriber types: psychologists, psychiatrists, and primary care physicians.

The discussion is written well, and I appreciated the limitations identified, particularly acknowledging that non-adherence could be mistaken for deprescribing in the current study.

Given the current landscape of polypharmacy, service provider shortages, and growing concern about long-term consequences of psychotropic medications, this manuscript is timely and contributes to the growing deprescribing literature. Recommend publication pending minor revisions. There are several errors to be fixed and a few areas requiring clarification, outlined below.

Abstract: 2 grammatical errors

• line 28: “used to developed” should be “used to develop”

• line 29: “from a psychologists, psychiatrists…” should be “from psychologists, psychiatrists…”

Line 89-90: Please either revise this statement or add more context: “In addition to the potential for these deprescribing barriers and facilitators to disproportionately impact PCPs, the opposite may be true for prescribing psychologists.” It is unclear from the preceding paragraph and text following this sentence how would the opposite impact of any barriers or facilitators would be true for psychologists. Please add more information.

Lines 105-106: This statement is presented as a given, but does not seem supported in the current text: “the potential for known deprescribing barriers to impact prescribers differently”. Please either revise or add text/citations to support this statement.

Lines 111-112: Suggest adding a citation to support the statement that long-term use of psychotropic medication could be warranted: “Finally, while deprescribing is considered a positive outcome in the context of this study, we also acknowledge that long-term medication usage may be the correct treatment decision for a given patient.”

Figures 1 & 2: Scanned quality of the figures is poor, ensure final submission includes higher quality graphics.

Line 157-158: Sentence needs revision to correct multiple grammatical errors: “Patients who do not experience deprescribing during the follow up were be considered censored at the end of the year or their last day of enrollment.”

Tables 1 and 2: The titles of both tables are identical “Hazard Ratios and 95% Confidence Intervals comparing Deprescribing Rates between Prescribing Psychologists and Psychiatrists” – I believe Table 2 refers to PCP, not Psychiatrists?

Line 317: Table 10 is referenced in error: “The doubly-robust model supported this (Table 10),…”

Line 326-327: Table 11 is referenced in error: “The model supported this (Table 11),…”

References: The authors are encouraged to format their references consistently according to the style of the journal – some of the cited titles have all words in upper case, and some only capitalize the first word of the journal title. Consistent formatting at the manuscript submission stage would reduce burden on the journal editing team.

Reviewer #2: First, I want to thank the group for taking on this important research. Deprescribing is a field that warrants further study and this work is very important to our patients and to help inform training and public policy efforts. Many thanks.

Overall, the paper is well written and concise.

I had a few questions about the methodology used:

1) (Line 165): How were inpatient services delineated for prescribing purposes. For example, if a patient was admitted to medicine with a psychiatry consult, was that patient then attributed to a PCP because the admitting attending was internal medicine? Clarification here would be helpful, because in these cases the subsequent PCP may be reluctant to deprescribe a medication started by a psychiatrist.

2) (Line 168): I would include some discussion here about how the possibility of a PCP starting a medication and simultaneously referring to psychiatry/psychology would be handled in this approach. Often wait times for specialist care may be quite long and this might lead claims to be attributed to PCPs that were actually managed by psychiatry/psychology. This would be particularly relevant if the the employer-based payers required PCP referral for specialist care rather than patients being able to access specialist care directly.

3) There is no mention of how advanced practice providers are categorized. Where I practice, there is a high prevalence of both psychiatric and family medicine/internal medicine nurse practitioners. Additionally, the prevalence of physician assistance is quite high. If these prescribers are included in any one of the cohorts, this should be made very clear in the explanation of how claims are attributed. These advanced practice providers are neither psychiatrists nor family medicine/internal medicine/pediatric physicians and may have dramatically different prescribing practices.

4) Within the limitations section, I would question whether the use of private, employer-sponsored insurance is a limitation since the results may not be generalizable to the Medicare/Medicaid populations.

5) Within the limitations section, recommend adding a caveat about fragmented care. In my own practice, refills from PCPs seem to be perpetuated even when I've assumed care of the patient as their psychiatrist and so it is possible that claims data do not reflect what the clinician currently managing the patient's mental health care believe the patient is/should be taking.

6) (Line 178): for the deprescribing sections, would clarify whether approach #3 (sustained reduction in prescribed days' supply) takes into account daily dose? In the section around line 178 it does not, but then line 361 refers to tapering, which typically takes a daily medication and systematically reduces the dose/day, rather than the days' supply.

7) (Line 349): question the assertion that high degree of deprescribing is a positive outcome of deprescribing. At least some of this may be related to patient/clinician unmet expectations of the benefits of psychotropic medications. Your assertion implies that the patient is doing better and therefore their clinician deprescribes, but a very different possibility remains that the patient is doing poorly on their medication and is tired of side effects so stops filling their prescriptions and continues to do poorly. Without outcomes data, this assertion is a stretch.

8) (Lines 380-388): This section is also a stretch. While you conclude that additional research is needed, I think the universally positive hypotheses listed detract from the overall paper and may lead your readers to discount your research due to perceived bias. Would recommend restating as a broader "More research is necessary to elucidate the mechanisms behind the lower rate of deprescribing among psychologists"...or something along those lines.

Overall, I enjoyed the paper very much and look forward to seeing it published and to seeing more research on this topic out of your group. Thank you.

**Do you want your identity to be public for this peer review?** For information about this choice, including consent withdrawal, please see our Privacy Policy

Reviewer #1: No

Reviewer #2: **Yes: ** Heather M. Wobbe, DO, MBA

---

## [Decision Letter · Decision Letter 1]

26 Nov 2025

Psychotropic deprescribing across different prescribing professions in New Mexico and Louisiana

PMEN-D-25-00326R1

Dear Dr. Hughes,

We are pleased to inform you that your manuscript 'Psychotropic deprescribing across different prescribing professions in New Mexico and Louisiana' has been provisionally accepted for publication in PLOS Mental Health.

Best regards,

Karli Montague-Cardoso

Staff Editor

PLOS Mental Health

Reviewer Comments (if any, and for reference):

Reviewer's Responses to Questions

**Comments to the Author**

Reviewer #2: All comments have been addressed

publication criteria?

Reviewer #2: Yes

3. Has the statistical analysis been performed appropriately and rigorously?

Reviewer #2: I don't know

4. Have the authors made all data underlying the findings in their manuscript fully available (please refer to the Data Availability Statement at the start of the manuscript PDF file)?

Reviewer #2: Yes

5. Is the manuscript presented in an intelligible fashion and written in standard English?

Reviewer #2: Yes

Reviewer #2: Thank you for your revisions.

**Do you want your identity to be public for this peer review?** For information about this choice, including consent withdrawal, please see our Privacy Policy

Reviewer #2: No
